# The Rehapiano—Detecting, Measuring, and Analyzing Action Tremor Using Strain Gauges

**DOI:** 10.3390/s20030663

**Published:** 2020-01-24

**Authors:** Norbert Ferenčík, Miroslav Jaščur, Marek Bundzel, Filippo Cavallo

**Affiliations:** 1Department of Cybernetics and Artificial Intelligence, Faculty of Electrical Engineering and Informatics, Technical University of Košice, Letná 9, 07602 Košice, Slovakia; norbert.ferencik@tuke.sk (N.F.); miroslav.jascur@tuke.sk (M.J.); 2The Biorobotics Institute, Scuola Superiore Sant’Anna, 560 25 Pisa, Italy; filippo.cavallo@santannapisa.it

**Keywords:** strain gauge, tremor quantification, Parkinson’s disease, action tremors

## Abstract

We have developed a device, the Rehapiano, for the fast and quantitative assessment of action tremor. It uses strain gauges to measure force exerted by individual fingers. This article verifies the device’s capability to measure and monitor the development of upper limb tremor. The Rehapiano uses a precision, 24-bit, analog-to-digital converter and an Arduino microcomputer to transfer raw data via a USB interface to a computer for processing, database storage, and evaluation. First, our experiments validated the device by measuring simulated tremors with known frequencies. Second, we created a measurement protocol, which we used to measure and compare healthy patients and patients with Parkinson’s disease. Finally, we evaluated the repeatability of a quantitative assessment. We verified our hypothesis that the Rehapiano is able to detect force changes, and our experimental results confirmed that our system is capable of measuring action tremor. The Rehapiano is also sensitive enough to enable the quantification of Parkinsonian tremors.

## 1. Introduction

Parkinson’s disease (PD) is among the most common neurological diseases. In 2016, approximately 6.1 million people worldwide were affected. This is a substantial increase compared to the 2.5 million PD patients reported in 1990. Furthermore, between four and 20 new cases per 100,000 people are reported annually [1]. Improved longevity and more precise diagnostic techniques for PD have both contributed to this increase [2].

PD is a chronic, progressive disease involving a gradual loss of motor and non-motor functions [3]. Non-motor symptoms include apathy, anhedonia, and depression. The loss of motor functions manifests as tremor, bradykinesia, rigidity, and a loss of postural reflexes [4]. Parkinsonian tremor (PT) is an approximately sinusoidal oscillatory motion that varies based on the severity of the disease, the activity performed by the affected individual, and their level of stress. Tremor may have a varying amplitude with a fixed frequency, which complicates its detection [5]. Because of the periodic nature of PT, Fourier transform (FT) is commonly used to analyze sensor data in the frequency domain [6]. Therefore, wearable sensors, for instance, inertial measurement units (IMUs) and electromyography (EMG), with subsequent spectral analysis are the standard approach in the evaluation of tremors [7,8,9,10].

The Rehapiano has strain gauges placed on ergonomically designed handles for both hands. These gauges measure the forces—both voluntary and involuntary—that the fingers apply to the gauges, with an appropriate sampling frequency and high precision. The Rehapiano is an alternative to tremor quantification using accelerometers, gyroscopes, and EMG. It does not require a sensor to be attached to the subject, which is certainly an advantage over wearable sensors. As the Rehapiano measures action tremor under specific conditions, wearable sensors have a significant advantage as they can monitor patients in different situations. The Rehapiano is applicable for evaluation/diagnosis and for fine motor function rehabilitation in clinical use. A quantitative assessment of a subject’s tremor serves as decision support for the physician who determines the dosage and the type of medication to be administered. Several studies suggest that there is a link between the severity of the PD tremor and the stage of PD [11].

The authors consider the contribution presented here to be as follows,
an introduction of the Rehapiano device for fast detection and quantification of action tremor using strain gauges,validation of the system by the comparison of measurements made by the Rehapiano to those made using optical encoders,an experimental analysis of the Rehapiano on healthy subjects and patients with PD, andan adaptation of an algorithm [9] that was previously developed for use with accelerometers and gyroscopes to asses tremor severity.

Section 4 describes the novelty of the device hardware. The methodology of measuring tremor with strain gauges is presented in Section 5 and Section 6. The results of our experiments support this unique approach towards evaluation of PT. Therefore, this framework could provide an alternative to standard approaches to PT assessment.

## 2. Problem Statement

Tremor is characterized in medicine as an involuntary rhythmic and periodic movement of body parts. All body parts may be affected, including the head, chin, and soft palate [12]. Muscle contractions during a tremor have a regular frequency [13]. However, identification of the frequency may be complicated by the signal’s amplitude changes. The amplitude changes may occur spontaneously, but are often correlated with change in the limb’s position, fatigue, or emotional stress [5]. Frequency is the tremor’s basic descriptive criterion, being categorized as low (<4 Hz), medium (4−7 Hz), or high (>7 Hz). The frequencies of tremors of different etiology have differential diagnostic value: cerebellar, Holmes, or palatal tremor have slow frequency, whereas orthopedic tremor is very rapid [14].

Tremor, according to its etiology, is categorized as rest tremor or action tremor, the latter being further subdivided into action postural and action kinetic tremor (see Figure 1) [15]. A rest tremor appears on relaxed muscles and should be measured on a lying subject. Action tremor appears on muscles that have been voluntarily engaged. A specific isometric tremor appears also in healthy subjects during strong isometric muscle contractions but can be superimposed with a tremor of another type [16].

From a clinical point of view, we recognize physiological, essential, Parkinsonian, and orthostatic tremor, tremor associated with another neurological disease, and psychogenic tremor [17]. Physiological tremor occurs in fine motor activities and is normal. It can be accentuated by anxiety, emotional stress, and some medications or by underlying conditions such as hypothermia, hypoglycemia, or hyperthyroidism. The frequency of a physiological tremor ranges from 4 Hz to 8 Hz (usually more than 7 Hz) [18]. Essential tremor is a sign of a health problem. It is characterized by the presence of bilateral and predominantly symmetrical, action postural or action kinetic, permanent and visible tremor, especially of the upper limbs. The essential tremor frequency is between 4 Hz and 10 Hz. More than 50% of affected subjects report a tremor reduction after consuming alcohol [14].

PT manifests itself in three types: resting, action postural, and action kinetic tremor [19]. Action tremor often has been observed in Parkinson’s disease (PD). The prevalence may be as high as 92% [20,21]. PT is often reduced during movement, although sometimes it is not, and then usually has the same frequency as the rest component. Typically, there is a pause in the tremor during a change from rest to posture [22]. PT frequency is usually above 5 Hz [23], although the upper frequency limit has not yet been established. Especially in the early stages of PD, the PT frequency ranges up to 8 Hz. Tremor is the most common initial symptom of PD, occurring in approximately 70% of Parkinson patients [24]. Its onset is usually one-sided. Some patients with clinical manifestations describe a strange feeling of internal shivering in the limb(s) or inside the body. Initially, the tremor occurs only intermittently from fatigue, agitation, or excessive concentration, and it disappears during sleep [25]. PT may resemble coin-counting or pill-rolling. The thumb moves with simultaneous flexion and extension of the metacarpophalangeal and interphalangeal joints [26]. During the course of the disease, the tremor may expand, and forearm and shoulder movements appear.

The primary orthostatic tremor manifests itself as a perceived sense of instability that reduces when walking. Tremor and the feeling of instability worsen during prolonged standing. Fine, high-frequency tremor (from 14 Hz to 16 Hz) may be felt by palpating the lower limbs rather than detected visually [27,28]. Tremor can be associated with a multitude of neurological diseases. Cerebellar tremor, Holmes tremor, and dystonic or neuropathic tremor have specific features, as described in [17]. Psychogenic tremor (a synonym for functional tremor) is the most common form (55%) of psychogenic motor disorders. Seventy-five percent of the affected people are female. A psychogenic tremor’s frequency is less than 7 Hz [29].

Tremor is a significant symptom and its quantification can aid in diagnosing the related problem, determining the right dosage and the right type of medication, and evaluating the development of the symptom over time. When measuring Parkinsonian, psychological, or essential tremor with existing devices, it is often difficult to repeat the exercise and the related measurement under the same conditions.

## 3. Background and Related Works

Determining subjects’ activity recognition and monitoring is important for understanding their condition and the development of the disease [8]. In Elble and McNames [30], a practical overview of the use of portable motion transducers in the quantification of tremor is provided. Rather than a comprehensive review of the transducers available for the assessment of tremor, it is a practical guide to the selection and use of portable transducers in tremor analysis. Elble and McNames determined what sensitivity, amplitude, and frequency ranges that transducers should use for high fidelity tremor detection. Tremor measuring devices can be divided into wearable sensors and fixed devices. The wearable sensors must be small, light, and securely affixed to the related body part.

In Haubenberger et al. [7], various types of devices for quantification and characterization of tremor are compared. The authors cover electromyography, accelerometry, gyroscopy, activity monitoring, digitizing tablets, and acoustic analysis of voice tremor. Availability on the market, ability to use, acceptability, reliability, and responsiveness were reviewed for each measurement method. The following criteria were adopted to evaluate each measurement method; (1) The use in the assessment of tremor, (2) use in published studies by people other than the developers, and (3) adequate clinical testing. Based on the criteria set out in this review, accelerometry, gyroscopy, electromyography, and tablet digitization met all three criteria for use in quantifying and detecting abnormal tremor. Some studies have indicated that accelerometer and gyroscope measurements correlate strongly with the unified PD rating scale (UPDRS) [31]. The United States Food and Drug Administration (FDA) approved the Kinesia™system (Great Lakes NeuroTechnologies Inc., Cleveland, OH, USA), which is used to assess Parkinsonian symptoms with an inertial measurement unit (IMU), which embeds a three-axis gyroscope and a three-axis accelerometer in a single chip on top of a finger [32]. In Niazmand et al. [33], a study about using a wireless wearable sensor system for evaluation of the severity of motor dysfunction in PD is presented. The system was integrated into a smart glove equipped with two touch sensors, two 3D-accelerometers, and a force sensor to assess the cardinal motor symptoms of PD (bradykinesia, tremor, and rigidity of the hand and arm). The study focused on the hardware, which includes a glove with a control and transmit unit, a receiver unit, and a computer for storage and analysis of the data. In Khan et al. [34], BioMotion Suite’s wearable system kit, equipped with a triaxial accelerometer, was used. The sensor samples at a rate of 32 Hz and has a range of ±3 g. Data are processed using proprietary BioMotion Suite software implemented in Matlab. Khan et al. processed their experimental data with six classification algorithms to classify PD data. Accelerometer data from measurements often produce noisy data, which complicates their processing.

## 4. The Rehapiano

The Rehapiano (see Figure 2) is a system that is comprised of dedicated hardware and analytic software that provides repeatable results, measures and quantifies the subjects’ tremor, and can monitor progression of their disease. The primary use of our system is to measure and analyze tremor based on measuring force applied by the fingers. The Rehapiano can also be used for the rehabilitation of fine motor skills. We plan a clinical trial that will include measuring the tremor of healthy subjects over a period of time and the tremor of subjects with PT after each cycle of medication. These measurements serve to determine the individual’s progress (improvement, deterioration, or stagnation), and in the case of the affected subjects, constitute decision support for further treatment.

The base of the device consists of an aluminum alloy frame 600 mm wide and 360 mm long. Aluminum beams have a square cross section of 30×30 mm^2^. The device weighs 16 kg and is stable when subjects work with it. Subjects place their forearms into low-temperature thermoplastic splints designed specially for the Rehapiano. The splints are equipped with three Velcro fasteners, each of which has a soft lining on the inside. The hands are in the plate from the wrist to the elbow in a fixed position.

The padded splints are hygienically harmless, non-allergic, and washable. They can be positioned and adjusted. Spacing is adjustable from 10 cm to 50 cm, tilt aside angle $∠120^{\circ }$, and ahead angle $∠40^{\circ }$. The ergonomic handles provide support for the palms and fingers and feature 5 strain gauges for each hand. The areas where subjects must put their fingertips are marked pink. The strain gauges measure the force applied by the subject’s fingertip in one axis. The splints that hold the forearm isolate the hand during the measurement. The strain gauges are powered by 5V and measure a maximum force of 50 N (5 kg). The strain gauges measuring the thumbs are placed on the vertical surface for increased ergonomy. PT frequency is usually above 5 Hz, [17,35]; therefore, we used an Hx711 (24 bit) analog-to-digital converter used in industrial control applications and sampled the measurements at 40 Hz [36].

The experimental data was acquired by an Arduino Mega and sent to a computer via USB interface (see Figure 3).

The measured values were sent together with the timestamps. The computer application was developed in C#. For easier transportation of the device, we used a minicomputer connected to the monitor. However, any monitor or projector can be used. The keyboard (see Figure 2) was used to enter the subject’s initials, date of birth, gender, and current health problems. This information was used for knowledge retrieval from the experimental data and was anonymized.

## 5. Methods

Three hypotheses were formulated:The Rehapiano is able to detect force changes with frequencies between 1 and 20 Hz.Rehapiano measurements can be used to detect tremors of Parkinson patients.The Rehapiano is sensitive enough to enable quantification of PT.

The first hypothesis was conducted for the evaluation of tremor simulation; the other two hypotheses involved human subjects. We conducted the experiments in laboratory conditions, on healthy and Parkinson subjects. Physicians of Parkinson patients reported to us that the PD patients also manifest action tremor. The facilities that enabled us to conduct these experiments were involved in the development of the Rehapiano. We have complied with all legal and legislative conditions, including GDPR patient protection. Each patient was personally acquainted with the use of their data from the device and signed an agreement permitting processing the medical data.

### 5.1. Verification of Hypothesis 1

To verify Hypothesis 1, a device (see Figure 4) that simulates force oscillations of varying frequency was constructed. L298N H bridge connected to an Arduino Uno and an RDO-37KE50G9A 12V DC motor (a 7.4 W output, a 0.17 mN torque, and a 400 rpm rotation speed, with a gearbox and gear ratio of 1:9) and an encoder. We used a pulley with an eccentrically mounted elastic rubber band. The other side of the elastic rubber was placed on the strain gauge. Because we could control the DC motor’s revolutions per second precisely, we were able to generate a regularly varying force at the strain gauge. We validated the Rehapiano at three frequencies between 1.5 and 7 Hz. This range was chosen with regard to the usual PT frequencies.

### 5.2. Verification of Hypothesis 2

We created a subject measurement protocol. The Rehapiano is suited for measuring action tremor produced during voluntary muscle contraction [37,38]. Each measured subject was prompted to apply force with a given finger. The amplitude of this force is displayed using vertical bars. By inducing targeted muscle contraction, the tremor may manifest and the device measures the force at 40 Hz frequency.

Thirty-six healthy volunteers (average age: 41.72y, 21 females, 15 males) and seven PD patients (average age: 76.10y, 6 females, 1 male) participated in this study. We also had a patients with leg tremors and dyskinesia, without any tremor in their hands, and were therefore not recruited. The PD subjects were evaluated also by the physician according to the Fahn–Tolosa–Marin Tremor Rating Scale (FTMTRS) (see Table 1), which quantifies rest, postural, and action/intention tremor. The Sessions column indicates the number of measurements for all fingers. Patient 3 had a higher tremor when he tried to write by hand, so he was rated 4 (unable to hold a pencil) by a physician. The FTMTRS is a widely used clinical rating scale quantifying severity of tremor from 0 (none) to 4 (severe) for the given body part [39,40].

The application shows which finger is presently being measured and the force value to be achieved. During the entire measurement, the patient is guided by a virtual nurse that is giving the subject instructions on what to do. When the subject reaches the desired value, the virtual nurse prompts the subject to maintain the force for a given time. If the measurement is valid, the application continues and the measurement is performed with the next finger. If not, the measurement is repeated. The measurement protocol can be modified by changing the required sequence of fingers, time, and force to be maintained. However, we used the same protocol parameters for all subjects.
The target value—set to 300 g—is the force produced by pressing on the strain gauge. All PD subjects could exert a finger force of 450 g on average. The experimental target value was set to 2/3 of 450 g—300 g for all fingers.The hold time—set to 3 s—is the period during which a patient should keep the force around the target value. This value was selected based on two aspects: we assumed that a longer exercise than 3 s for all fingers would lead to fatigue and that a shorter exercise would not contain enough data to evaluate the tremor.The sequence of fingers represents the sequence of fingers without repetition from the left little finger to the right little finger. In these experiments, we chose the most simple sequence to iterate through the fingers from left to right to make the exercise as simple as possible.

The example in Figure 5 shows the measurement of the subject patient’s left hand with the middle finger. The subject does not see the target value.

### 5.3. Verification of Hypothesis 3

To determine whether the Rehapiano is sensitive enough to quantify the frequency of tremor on both hands of a patient, we measured a PD patient with both sides of the body affected. The whole exercise was performed under the supervision of a physician. For health reasons, the subject was not under the influence of medication. The subject also suffered from pain in the right shoulder and lowered fine motor skills in the right arm, as compared to the left arm. We repeated the aforementioned measurement protocol seven times for both hands with the hold time extended from 3 to 5 s for each finger, with the target value and sequence of fingers remaining the same. We acquired 28 measurements for each hand, excluding the thumbs. Based on the methodology presented in [9], we evaluated the validity of the measurements based on the peak power proportion. The ratio of the area below the power spectral density (PSD) curve around the detected dominant frequency (±0.3 Hz) and the whole area below the curve had to be higher than a set threshold to consider the measurement valid; see Figure 6, Equation (Equation 2).

Equation (Equation 1) shows how the peak power from Figure 6 is calculated:(1)$$P_{\mathrm{peak}}=\int_{f_{1}}^{f_{2}}\frac{FT^{*}\left(\;\mathrm{signals}\;\right)\times \mathrm{FT}\;\left(\mathrm{signals}\right)\;}{N^{2}}df,$$
where FT* is the FT conjugate, and *N* is the number of received samples of the signal.

The peak power proportion $V_{f}$ is calculated as
(2)$$V_{f}=\frac{P_{peak}}{\sum P_{i}},$$
where $P_{i}$ is the power estimation of the specific frequency, and $\sum P_{i}$ is the sum over all powers in a frequency domain.

## 6. Experimental Results

### 6.1. Validation of the Rehapiano

Hypothesis:$H_{0}:\mu_{oe}=\mu_{r}$: Mean frequency measured by the optical encoder and by the Rehapiano is equal.$H_{a}:\mu_{oe}\neq \mu_{r}$: Mean frequency measured by optical encoder and by the Rehapiano is not equal.

We simulated tremors at three frequencies: low (≈1.5 Hz), medium (≈3 Hz), and high (≈7 Hz). For each frequency, we conducted 40 measurements, each 3 s long. Both the optical encoder data and strain gauge signal were recorded, and we calculated the frequency from optical encoder measurements for each 10 s measurement. We then used Fourier transform on the signal recorded by the Rehapiano. The resulting frequency is the frequency at maximal value of the single side power spectrum. The results of these measurements for high frequency are shown in Figure 7.

Subsequently, we calculated the mean and the variance of the measurements and from these values we computed the z-test. The z-test verifies our hypothesis that the measurement means of the Rehapiano and the optical encoder are the same at a significance level α=0.05. Table 2 shows the results of the hypothesis test for three frequencies. We cannot reject the null hypothesis for medium and high frequency at the 5% significance level. Therefore, we accept the null hypothesis for medium and high frequency, that $\mu_{oe}=\mu_{r}$. However, we reject the null hypothesis for low frequency $\mu_{oe}\neq \mu_{r}$.

Because we did not succeed in verifying the Rehapiano for measuring the tremor frequencies lower than 3 Hz, in the following experiments, we considered only measurements of tremor frequencies between 3.5 and 7.5 Hz [9]. In this range, we considered the Rehapiano measurements to be valid for the purpose of detecting and evaluating PT and considered Hypothesis 1 to be proven.

We are confident that the Rehapiano produces valid measurements for lower frequencies and that the failure to verify it lies with the verification device we have used. More on this topic is in Section 7.

### 6.2. Distinction between Healthy Population and Patients with PT

Hypothesis:Measurements from the Rehapiano contain detectable tremor information. The performance metrics of a classifier that detects tremor should meet the following requirements.
−Cross validation accuracy > 90%−Precision > 95%−Recall > 95%

We collected 490 measurements of 43 subjects from 49 sessions. Of those 43 subjects, 36 were healthy and seven were PD patients, with tremor rated on the FTMTRS (Table 1). All the healthy subjects completed the session once, producing 10 measurements each. Two PD patients repeated the exercise more than once (Table 1). Ninety-eight thumb measurements were uniformly excluded from the dataset, because of the PD patients’ inability to maintain pressure on thumbs. We created the dataset with the following pipeline.
First, our algorithm filters the raw signal with an outlier filter that replaces values below the 1.25th percentile of the distribution using linear interpolation.It then applies a band-pass filter that keeps frequencies between 3.5 Hz and 7.5 Hz.Next, the algorithm calculates a one-sided amplitude spectrum of a 3 s signal, where the patient reached the desired force (Figure 8). After that, it resamples the result of the FT at 0.01 Hz between 3.5 and 7.5 Hz, creating a vector with 41 values describing the FT amplitude of the signal at specific frequencies.Finally, we expect that FT amplitudes of the PD patients will be significantly different from the healthy population, and the data are labeled based on this assumption (PD patient: 1; Healthy Subject: 0).

We trained four binary classifiers—support vector machines (SVM), naive Bayes (NB), decision tree (DT), and K-nearest neighbors (KNN). The target class of the classification was patients with PD. Classifiers had the following parameters: SVM: Gaussian kernel with γ=6.4; NB: nonparametric Gaussian NB; DT: maximal number of nodes was set to 100, and the optimal feature for current node was selected by Gini’s diversity index; KNN: k=13, and the distance metric was cosine similarity. Input features of KNN and SVM were standardized. The mean of the feature was subtracted from every value in the column, and this value was divided by its standard deviation. We validated the classifiers with fivefold cross-validation; the cross-validation accuracy is shown in Table 3.

Table 4 includes five metrics. Validation accuracy represents the average accuracy of classification from every K-fold classifier. Sensitivity shows the ability of the classifier to detect PD patients that truly have PT, whereas specificity is the ability to identify healthy people that do not have PT. Precision is the probability of making correct decision, when our classifier categorizes the measurement as a patient with PT. Subsequently, the $F_{1}$-score is the weighted average precision and sensitivity, and the formula is $F_{1}=2*sensitivity*precision/\left(sensitivity+precision\right)$. All classifiers reach a validation accuracy higher than 90%. Although only the DT reaches sensitivity values higher than 95%, all four classifiers have a specificity higher than 95%. We highlight the DT that has the best F1 score.

### 6.3. Quantitative Assessment of the Tremor

Hypothesis:Measurements from the Rehapiano provide quantitative information about tremor. Subsequent measurements of the same subject output the same tremor frequency.
−Standard deviation of the measurements is less than 0.15 Hz.

We collected 48 measurements from one subject with PD. We repeated the first two steps from the data processing pipeline as described above. Subsequently, we applied Fourier transform on the signal and calculated the dominant frequency and peak power proportion from the power spectral density.

Table 5 contains the results of repeated measurements of a PD subject. Measurements are valid if the value of peak power proportion (Equation (Equation 2)) of the measurement is higher than the peak power proportion threshold. The peak power proportion threshold is a value above which we do not include measurement into the calculation of tremor frequency. We experimented with five peak power proportion thresholds from 0.5 to 0.9. The table is split into measurements of the right and the left hand. For each hand at a specific threshold, we provide the following information; the relative number of valid measurements, the absolute number of valid measurements, and the dominant frequency and its standard deviation. Absolute and relative number (absolute /total) of valid measurements both describe the amount of successful trials. The standard deviation of the measured PD tremor frequency for the right hand increased with the power peak ratio threshold, and we could obtain only two valid measurements at the threshold equal to 0.7. Because this was too small a statistical sample, we have not considered these measurements. We obtained nine valid measurements for the left hand even at the increased threshold equal to 0.8. The results indicate that the Rehapiano can be used to perform repeatable measurements. The mean of the calculated PD tremor frequency was 6.93 Hz with a standard deviation below 0.1 Hz.

## 7. Discussion

We developed the Rehapiano device for the quick evaluation of action tremor. Tremor is measured during a steady state, when fingers are exerting the desired force. From a clinical viewpoint, action tremor, specifically kinetic (see Figure 1), appears during the targeted motion. Such movement is represented many times during everyday activities. Therefore, we decided to develop a device that measures this tremor, and based on the results of a particular activity, a physician can observe the development of PT.

Concerning Hypothesis 1, we rejected the null hypothesis based on the low frequency hypothesis test. Based on our observations, we still have confidence that the Rehapiano is capable of reliably measuring tremors of lower frequencies and that the problem of verifying it lies in the DC motor and the rubber band used for its verification. The difference between the measurements from the encoder and the Rehapiano may be due to the low torque of the DC motor at low voltage. The rubber band slows down the motor too much when being stretched and then, when being contracted, causes a peak in its angular velocity. This may distort the measurement. The authors will build a new verification device and repeat the procedure in the near future. For medium- and high-frequency tremors, we have confirmed Hypothesis 1.

Concerning Hypothesis 2, two factors influence the quality of the measurements and the resulting accuracy of the classifiers. The first factor is the subject’s stress. According to the work in [16,45], the amplitude and frequency of tremor under psychological stress conditions increase significantly compared to a calm state. The second factor is the medication the subject receives. Based on the subjective opinions expressed by the subjects, they did not feel stressed during the measurement. All PD-affected subjects (except for the one measured in relation to Hypothesis 3) were taking medication to relieve PD symptoms. All the subjects completed the measurement protocol successfully. Two subjects had to repeat the measurement routine for one of their fingers due to wrong placement of the fingertip on the strain gauge. Based on the experimental results, we consider Hypothesis 2 to be confirmed.

As for Hypothesis 3, we subsequently measured both hands of a PD patient seven times. PT most often presents unilaterally and later progresses to include both sides of the body [46]. Both upper limbs of our subject were affected. Although the measurements of the left hand were valid, we had to ignore the measurements of the right hand. The patient was unable to keep the fingertips of the right hand in steady contact with the sensor, tapping the strain gauge more or less. From our perspective, this involuntary motion invalidated most of the right hand measurements. We discovered that, on multiple occasions, the signal contained several dominant frequencies. Therefore, we are currently working on several design and technological improvements to our Rehapiano device, primarily to improve the contact between the finger and strain gauge using neodymium magnets and gloves. Based on the experimental results of the PD subject’s left hand measurements, we consider Hypothesis 3 to be confirmed.

## 8. Conclusions

This paper presents the Rehapiano system for measuring force applied by the fingertips, used here to detect and quantify hand tremors. We focused on measuring the action hand tremor of healthy subjects and PD patients in this study. We can confirm that the Rehapiano is capable of measuring tremors with frequencies higher than 3 Hz, that the Rehapiano measurements contain detectable and useful tremor information, and that the Rehapiano produces repeatable results. We have implemented a measurement protocol aided by a virtual nurse. Our device is noninvasive and non-wearable. It differs from the existing solutions due to its low measurement time, its comfort level, and the rapid processing and evaluation of the results. The average time of the measurement procedure was three minutes, and the subjects were not fatigued. We will be replacing the Hx711 converters to increase the sampling frequency, thus obtaining finer and higher quality data, which will also enable measurement of orthostatic tremor (between 14 Hz and 16 Hz).

We plan to expand the sample of the test subjects and to create measurement protocols for patients of specific groups. Future work also includes using the Rehapiano as a hand fine motor rehabilitation device. We are developing a rehabilitation computer game with the Rehapiano as the game controller. The game should motivate patients with motor disorders to exercise and improve their fine motor skills. Based on the field work and cooperation with the medical team, we learned that no PD patient barring one had problems in using the Rehapiano. The strain gauges’ pods are equipped with neodymium magnets, although we have not yet used this feature. We plan to use fitting gloves with magnetic inserts to help keep the fingertips in place during the interaction. We have also learned that quantifying the tremor of PD patients at regular intervals could aid in determining the proper medication for them and in adjusting the dosage.

After carrying out all the planned improvements to the Rehapiano, we plan to use it to detect other types of tremor in clinical settings. Further, we want to compare Rehapiano measurements with reference measurements from IMUs and EMG. These experiments should provide conclusive evidence about the clinical feasibility of the Rehapiano.

## Figures and Tables

**Figure 1 sensors-20-00663-f001:**
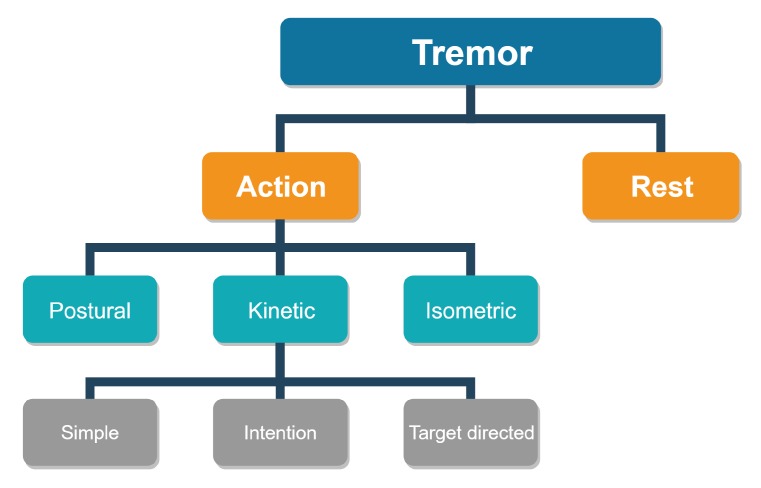
Basic division of tremors.

**Figure 2 sensors-20-00663-f002:**
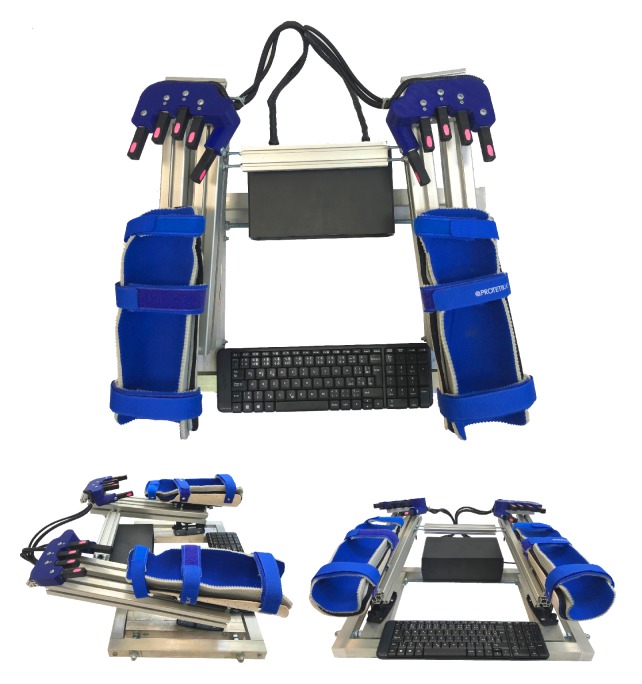
The Rehapiano device.

**Figure 3 sensors-20-00663-f003:**
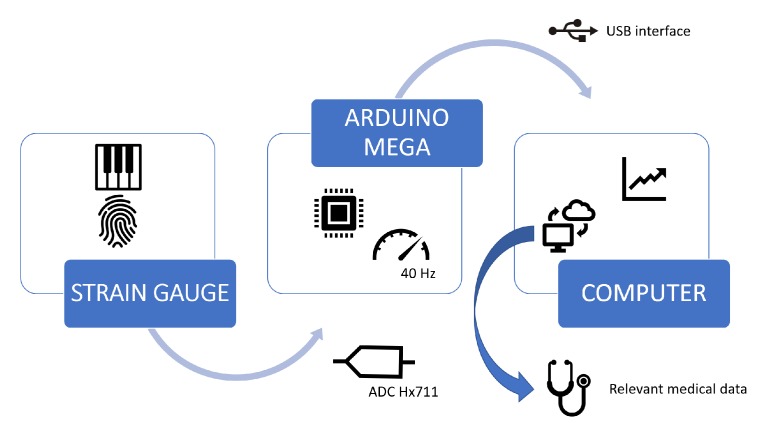
Rehapiano device data flow.

**Figure 4 sensors-20-00663-f004:**
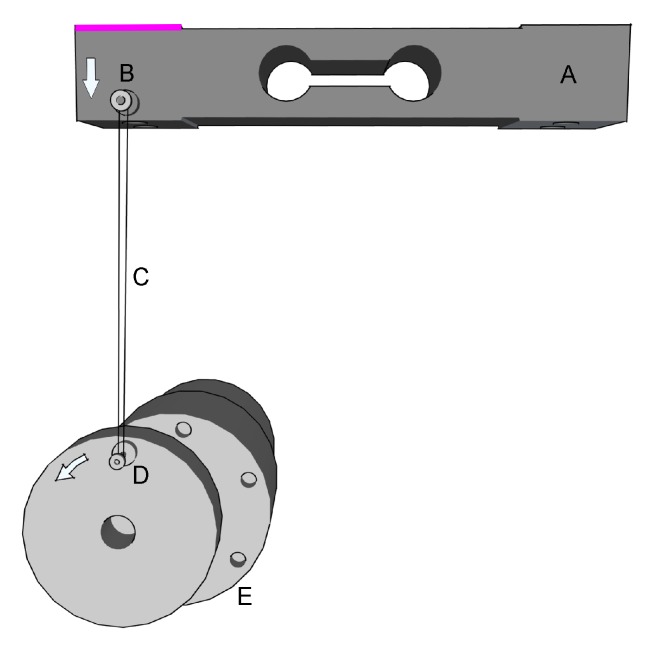
Experimental setup for Rehapiano validation. The direction of rotation of the DC motor and the force applied to the strain gauge are indicated by arrows. (A: fixed, B: force applied, C: elastic band, D: pulley, E: DC motor).

**Figure 5 sensors-20-00663-f005:**
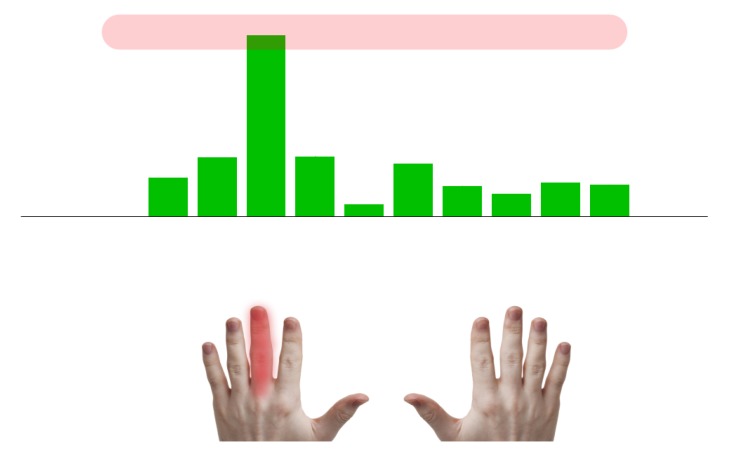
Example of finger force measurement. The required value is in the red area. The measured finger is highlighted and corresponds to the third vertical green bar from the left, and the y-axis represents the applied force.

**Figure 6 sensors-20-00663-f006:**
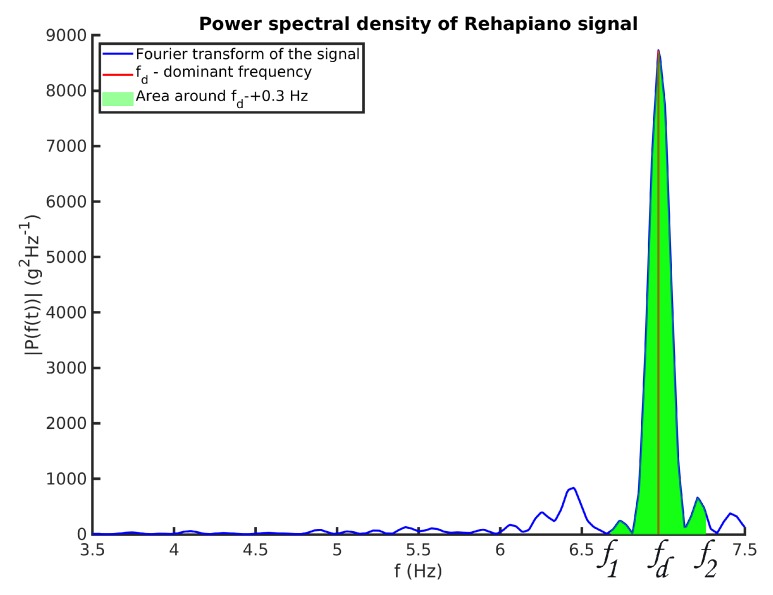
Power spectral density (PSD) of a PD patient measurement of a single finger. $f_{d}$ represents the dominant frequency of the signal.

**Figure 7 sensors-20-00663-f007:**
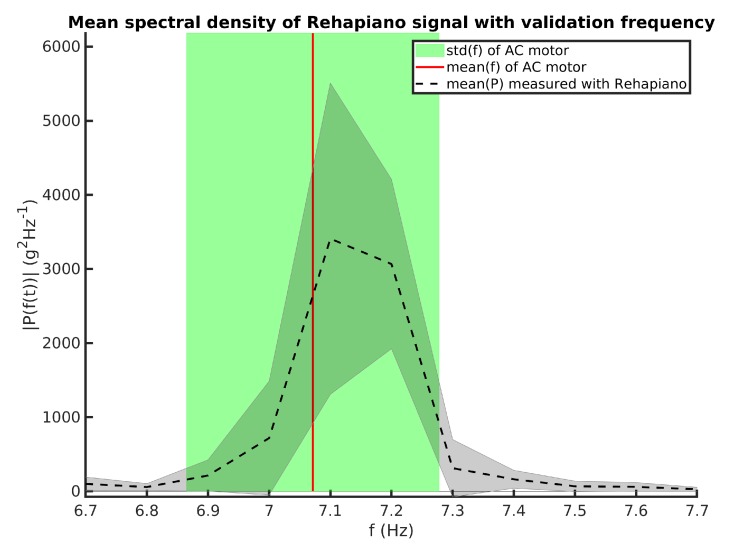
Mean frequency measured with optical encoder (red line), its standard deviation (green area), the Rehapiano mean power spectral density (black line), and its standard deviation (gray area) for high frequency experiments.

**Figure 8 sensors-20-00663-f008:**
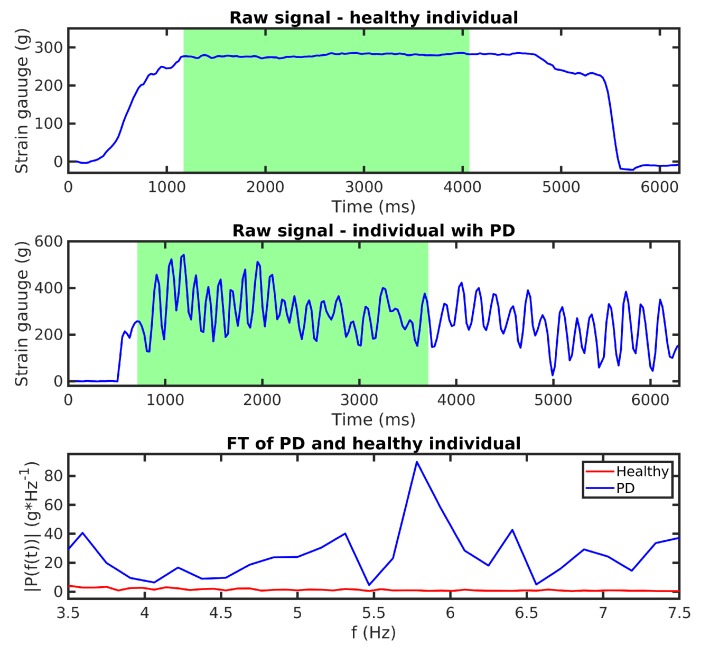
Comparison of healthy individual exercise and exercise for a patient with PD. Resampled values of FT transform (3rd) are used as input to our classifiers.

**Table 1 sensors-20-00663-t001:** Fahn–Tolosa–Marin tremor rating scale values of the participants.

PD Patient	Tremor	Handwriting	Sessions
Patient 1.	1	2	2
Patient 2.	2	3	1
Patient 3.	3	3	1
Patient 4.	3	4	1
Patient 5.	3	2	7
Patient 6.	3	2	1
Patient 7.	4	3	1

**Table 2 sensors-20-00663-t002:** Mean frequencies measured by optical encoders and the Rehapiano and their variance, the result of the z-test, and the decision as to whether we can reject the null hypothesis.

Freq	$\mu_{\mathrm{oe}}\left(\mathrm{Hz}\right)$	$\sigma_{c}$	$\mu_{r}\left(\mathrm{Hz}\right)$	$s_{r}$	$z_{\mathrm{test}}$	$H_{0}$
Low	1.56	0.17	1.42	0.053	−2.51	1
Medium	2.96	0.09	2.94	0.021	−0.97	0
High	7.07	0.2	7.16	0.028	1.28	0

**Table 3 sensors-20-00663-t003:** Performance measures of different classifiers used to distinguish the healthy population from PD patients.

Class.	ValAccuracy	Sensitivity	Specificity	Precision	$F_{1}$
SVM	0.9311	0.798	0.9965	0.9965	0.8863
NB	0.9464	0.875	0.9722	0.9557	0.9136
**DT**	**0.9638**	**0.975**	**0.9861**	**0.9872**	**0.9811**
KNN	0.9285	0.9326	0.9756	0.9872	0.9537

**Table 4 sensors-20-00663-t004:** Comparison of different classifiers approaches on PT (LSTM: long short-term memory; GTB: gradient tree boosting; BCT: bagged classification tree; RF: random forest).

Name	Device	Scale	Method	Type	Accuracy	Sensitivity	Specificity
Our	Rehapiano	FTMTRS	DT	Binary	0.9638	0.975	0.9861
[41]	Gyroscope + Acc	UPDRS	LSTM/GTB	Multi	0.84/0.96 *	-	-
[42]	Leap Motion	UPDRS	BCT	Binary	0.99	0.99	0.99
[43]	Smartphone	UPDRS	RF	Binary	-	0.90	0.82
[44]	Accelerometers	Binary	Welch (2)	Binary	0.95	0.98	0.69

* Correlation to UPDRS.

**Table 5 sensors-20-00663-t005:** Results of repeated measurements with the Rehapiano of a PD subject.

	Left Hand Measurements			Right Hand Measurements		
$V_{f}\;\mathrm{threshold}$	Valid rel	Valid abs	f±std(Hz)	Valid rel	Valid abs	f±std(Hz)
0.5	0.8571	24	6.7954±0.4488	0.3928	11	6.27±0.3193
0.6	0.6071	17	6.8688±0.1471	0.2857	8	6.255±0.3792
0.7	0.4285	12	6.8925±0.126	0.0714	2	5.915±0.7566
0.8	0.3214	9	6.9344±0.0948	0	0	-
0.9	0	0	-	0	0	-

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
