# Peer review of "The Rehapiano—Detecting, Measuring, and Analyzing Action Tremor Using Strain Gauges"

_sensors, 2020, doi:10.3390/s20030663_

Round 1
Reviewer 1 Report
The authors introduced an interesting device (i.e. Rehapiano) to measure and analyze tremor of Parkinson’s patients. The device’s name suggests rehabilitation using the fingertips. However, the manuscript does not treat about rehabilitation, it only suggests (at the very end) that it could possibly be used in the future for this purpose.
The manuscript should be improved, several small mistakes can be found, and some sentences are hard to be understood. A few examples:
Line 73: What is the meaning of “/” in “Parkinsonian tremor is resting or action postural / action kinetic.”? If it is divided, simply write extensively.
Line 99: “of the the disease”
Line 135: “affected subjects”? Do you mean subjects diagnostic with Parkinson?
Line 139: “30 x 30 mm”; my suggestion: 30 mm x 30 mm or 30 x 30 mm²
Line 153: “by Arduino Mega”; my suggestion: by an Arduino Mega
Line 165: “was conducted by the evaluation”; my suggestion: was conducted for the evaluation
Line 173: “used the L298N H bridge connected to Arduino Uno and RDO-37KE50G9A”; my suggestion: used a L298N H bridge connected to an Arduino Uno and an RDO-37KE50G9A
Line 174: “(output 7.4W, torque 0.17Nm, rotation speed 400rpm”; my suggestion: (7.4 W output, 0.17 Nm torque, 400 rpm rotation speed
Line 199: “into the strain gauge”; my suggestion: on the strain gauge
In my opinion, the “we” is used way too often in the manuscript. I would suggest the authors to rewrite some sentences to avoid using “we” and to replace sometimes by the researchers or the investigators.
Another problem is the inconsistence of the authors. Paragraphs were normally started with space, but this was not the case in lines 165, 173, 182. Normally, the value and units were written in italic and without space. Very often there was a space in between (see lines 76 and 77), not in italic. Moreover, ranges were written in the ways (see lines 68 and 71). Abbreviations written with points at the end, others without. Sometimes these abbreviations are also not clear (an example is line 68: “app.”).
Several points were mentioned but not explained, results often not discussed. A few examples are:
Line 187: “Subjects with leg tremors or dyskinesia were not recruited”; why not?
Table 1: only patient 3 has larger handwriting tremor than his/her tremor itself. Any reasons for that?
Figure 5: Is the red area supposed to be the overlapping of the third bar with the gray chamfered rectangle at the top? Does the y-axis represent applied force? The fig. name says "middle finger force measurement" but there are measurements from all the others.
Table 2: How was the z-test obtained? This is not clear in the manuscript.
Line 255: “328 valid measurements”; how about the ones which were not valid? Why were they invalid? How many were they?
Line 270: add the meaning of F1 score
Table 4: Is F supposed to be frequency? If yes, it should be written as f. Moreover, is seems that amount of number after the comma could not be correct.
Table 4: Threshold of what?
Regarding hypotheses 3, it would be interesting to have another subject measured for a better evaluation.
Please add references in the text for “A psychogenic tremor’s frequency is less than 7Hz” and “Hx711 (24 − bit) analog-to-digital converter used in industrial control applications”.
The reference list has also several inconsistences. Most journals had their full names written, but this was not the case for reference 2. I have never seen “others” written in a publication; I would suggest using “et al.”. “pp.” written sometimes prior to page range and omitted for other references. Reference 20 had the journal paper written twice.
Author Response
The authors introduced an interesting device (i.e. Rehapiano) to measure and analyze tremor of Parkinson’s patients. The device’s name suggests rehabilitation using the fingertips. However, the manuscript does not treat about rehabilitation, it only suggests (at the very end) that it could possibly be used in the future for this purpose.
The manuscript should be improved, several small mistakes can be found, and some sentences are hard to be understood. A few examples:
Point 1: Line 73: What is the meaning of “/” in “Parkinsonian tremor is resting or action postural / action kinetic.”? If it is divided, simply write extensively.
Response 1: We have replaced Figure 1 with one that summarises the division of tremor more clearly and we cite the publication that clarifies this division.
Point 2: Line 99: “of the the disease”
Response 2: Extra “the” was deleted.
Point 3: Line 135: “affected subjects”? Do you mean subjects diagnostic with Parkinson?
Response 3: “affected subjects” was replaced with “subjects with PT” (PT stands for Parkinson tremor).
Point 4: Line 139: “30 x 30 mm”; my suggestion: 30 mm x 30 mm or 30 x 30 mm²
Response 4: We have changed millimeters to square millimeters.
Point 5: Line 153: “by Arduino Mega”; my suggestion: by an Arduino Mega
Response 5: We have added “an”.
Point 6: Line 165: “was conducted by the evaluation”; my suggestion: was conducted for the evaluation
Response 6: We have modified the form of the sentence.
Point 7: Line 173: “used the L298N H bridge connected to Arduino Uno and RDO-37KE50G9A”; my suggestion: used a L298N H bridge connected to an Arduino Uno and an RDO-37KE50G9A
Response 7: We have modified the sentence accordingly.
Point 8: Line 174: “(output 7.4W, torque 0.17Nm, rotation speed 400rpm”; my suggestion: (7.4 W output, 0.17 Nm torque, 400 rpm rotation speed
Response 8: We have modified the sentence accordingly.
Point 9: Line 199: “into the strain gauge”; my suggestion: on the strain gauge
Response 9: We have modified the sentence accordingly.
Point 10: In my opinion, the “we” is used way too often in the manuscript. I would suggest the authors to rewrite some sentences to avoid using “we” and to replace sometimes by the researchers or the investigators.
Response 10: We rephrased the paragraphs in which the first plural was and was often repeated in the whole manuscript.
Point 11: Another problem is the inconsistence of the authors. Paragraphs were normally started with space, but this was not the case in lines 165, 173, 182. Normally, the value and units were written in italic and without space. Very often there was a space in between (see lines 76 and 77), not in italic. Moreover, ranges were written in the ways (see lines 68 and 71). Abbreviations written with points at the end, others without. Sometimes these abbreviations are also not clear (an example is line 68: “app.”).
Response 11: We have corrected the inconsistencies, please refer to our manuscript with highlighted modifications.
Several points were mentioned but not explained, results often not discussed. A few examples are:
Point 12: Line 187: “Subjects with leg tremors or dyskinesia were not recruited”; why not?
Response 12: We have added the explanation why not. These subjects did not manifest tremor of upper limbs.
Point 13: Table 1: only patient 3 has larger handwriting tremor than his/her tremor itself. Any reasons for that?
Response 13: We have explained why. This patient was not able to hold a pen and the physician has evaluated his tremor as such.
Point 14: Figure 5: Is the red area supposed to be the overlapping of the third bar with the gray chamfered rectangle at the top? Does the y-axis represent applied force? The fig. name says "middle finger force measurement" but there are measurements from all the others.
Response 14: We've modified the caption accordingly. Indeed, all fingers are measured just the subject applies force by the middle finger upon the request..
Point 15: Table 2: How was the z-test obtained? This is not clear in the manuscript.
Response 15: We have added clarification to the manuscript.
Point 16: Line 255: “328 valid measurements”; how about the ones which were not valid? Why were they invalid? How many were they?
Response 16: We have added a measurement of a new PD patient with starting PD. We have added explanations on how certain measurements were excluded from the particular dataset. For example, we have excluded thumb measurements because some PD patients were not able to maintain the contact of the thumb and the strain gauge. We need to ad a fixture to Rehapiano to avoid this problem.
Point 17: Line 270: add the meaning of F1 score
Response 17: We have added the explanation of F1 score below Table 3.
Point 18: Table 4: Is F supposed to be frequency? If yes, it should be written as f. Moreover, is seems that amount of number after the comma could not be correct.
Response 18: We have changed it accordingly.
Point 19: Table 4: Threshold of what?
Response 19: We have added the explanation of peak power proportion threshold below Table 5.
Point 20: Regarding hypotheses 3, it would be interesting to have another subject measured for a better evaluation.
Response 20: We agree, however the methodology of hypothesis 3 requires several repeated Rehapiano measurements. We have recruited a new patient and obtained 2 valid measurements. This was sufficient for hypothesis 2 but insufficient for hypothesis 3. We will recruit a wider range of subjects in the near future.
Point 21: Please add references in the text for “A psychogenic tremor’s frequency is less than 7Hz” and “Hx711 (24 − bit) analog-to-digital converter used in industrial control applications”.
Response 21: We have added both references to converters and to psychogenic tremors.
Point 22: The reference list has also several inconsistencies. Most journals had their full names written, but this was not the case for reference 2. I have never seen “others” written in a publication; I would suggest using “et al.”. “pp.” written sometimes prior to page range and omitted for other references. Reference 20 had the journal paper written twice.
Response 22: We have modified the references.
Dear reviewer,
We are very grateful for your help in correcting our manuscript. We tried to incorporate almost all the comments you sent us. The device’s name suggests rehabilitation because in the future we want to create a rehabilitation game to be played with Rehapiano. Currently Rehapiano serves more for diagnostics than for rehabilitation. We agree that hypothesis 3 would have a greater value with another patient. However, other patients disagreed with so many repeated measurements. The number of patients who participated in the Rehapiano measurements were patients with Parkinson's disease who were currently in the hospital and agreed to participate. Nevertheless, we added a patient that we measured with tremor 1 and 2. In the future we want to increase the number of measured as much as possible. In this article we tried to prove the functionality of the rehapiano, so we think the number of patients was sufficient.
General remark: The English of the manuscript was corrected before submission. After our major revision, we will let it be corrected again by the MDPI English Editing Service. We include the revised manuscript and the differential manuscript in which changes are highlighted in the attachment.

Reviewer 2 Report
The authors have presented the device in descussion well, some improvements could be made to certain cited tremor frequncy data ranges, citing a study which validates claimed frequency ranges. However, the focus on the use and implementation of strain gauge sensors to detect tremors is not visible. I would suggest to either restructure the paper with focus on the technical sensor aspect of the work, or find another journal with health measurement and improvement focus.
Author Response
The authors have presented the device in discussion well, some improvements could be made to certain cited tremor frequency data ranges, citing a study which validates claimed frequency ranges. However, the focus on the use and implementation of strain gauge sensors to detect tremors is not visible. I would suggest to either restructure the paper with focus on the technical sensor aspect of the work, or find another journal with health measurement and improvement focus.
Response 1: Dear reviewer,
We are very grateful for your help in correcting our manuscript. We try to show the possibility of detecting tremor in chapters 4 and 5. We have submitted the paper to Sensors, because we have noticed that the following publications on tremor diagnostics were published by the journal:
Wearable Sensors for Estimation of Parkinsonian Tremor Severity during Free Body Movements by Murtadha D. Hssayeni ,Joohi Jimenez-Shahed ,Michelle A. Burack andBehnaz Ghoraani,
Automatic Classification of Tremor Severity in Parkinson’s Disease Using a Wearable Device by Hyoseon Jeon ,Woongwoo Lee ,Hyeyoung Park ,Hong Ji Lee ,Sang Kyong Kim ,Han Byul Kim ,Beomseok Jeon and
Support System to Improve Reading Activity in Parkinson’s Disease and Essential Tremor Patients by Franklin Parrales Bravo ,Alberto A. Del Barrio García ,Mercedes Gallego de la Sacristana ,Lydia López Manzanares ,José Vivancos andJosé Luis Ayala Rodrigo.
General remark: The English of the manuscript was corrected before submission. After our major revision, we will let it be corrected again by the MDPI English Editing Service. We include the revised manuscript and the differential manuscript in which changes are highlighted in the attachment.

Reviewer 3 Report
The paper is fine, minor issues. I suggest the following:
Line 45, change chapters to sections.
Line 165, where there only three experiments? It looks like you did many more, but the way it is written looks like there were only 3 experiments. Perhaps a different word?
Line 183, "The measured subject" should be "Each measured subject".
Line 184, how is force visualized on screen?
Line 242, drop either % or percent.
The same section, it is unclear whether your hypothesis was proven or disproven. Please clarify.
Line 255, explain why there were 36 healthy subjects (which look like control group) while only 6 real patients. At most, should they not be equal, or perhaps more patients?Line 271, two out of how many? Nine?
Discussion and conclusion: I am not a physician, nor knowledgeable in medical issues. Therefore, I cannot judge or comment on the issues discussed. Sorry.
Hope this helps.
Author Response
The paper is fine, minor issues. I suggest the following:
Point 1: Line 45, change chapters to sections.
Response 1: We have changed chapters to sections.
Point 2: Line 165, where there only three experiments? It looks like you did many more, but the way it is written looks like there were only 3 experiments. Perhaps a different word?
Response 2: We have changed the meaning of the sentence so that it didn't look like we did only three experiments.
Point 3: Line 183, "The measured subject" should be "Each measured subject".
Response 3: We have added “each” to the beginning of the sentence.
Point 4: Line 184, how is force visualized on screen?
Response 4: We have added a description of how force is expressed.
Point 5: Line 242, drop either % or percent.
Response 5: We left only a percent sign.
Point 6: The same section, it is unclear whether your hypothesis was proven or disproven. Please clarify.
Response 6: At the end of the hypothesis we added that it was proven. We also rephrased the section on confirming hypothesis 1.
Point 7: Line 255, explain why there were 36 healthy subjects (which look like control group) while only 6 real patients. At most, should they not be equal, or perhaps more patients?
Response 7: The number of patients who participated in the Rehapiano measurements were recruited from the currently available subjects in the hospital that have agreed to participate. We have added one more patient to the study with tremor 1 and 2. We plan to do a larger scale study. We have tried to prove that the concept of the Rehapiano is valid, and we think the number of patients was sufficient for this purpose at this point.
Point 8: Line 271, two out of how many? Nine?
Response 8: The total number was four and we have modified our sentence to make this clear.
Discussion and conclusion: I am not a physician, nor knowledgeable in medical issues. Therefore, I cannot judge or comment on the issues discussed. Sorry.
Hope this helps.
Dear reviewer,
We are very grateful for your help in correcting our manuscript. We tried to incorporate all the comments you sent us.
General remark: The English of the manuscript was corrected before submission. After our major revision, we will let it be corrected again by the MDPI English Editing Service. We include the revised manuscript and the differential manuscript in which changes are highlighted in the attachment.

Reviewer 4 Report
The paper presents a novel device to identify the action tremor in people with PD using FTMTRS. The algorithm for PD detection is interesting, however I notice some flaws in their experimental tests and important technical errors that must be corrected.
Minor Changes:
The paper is well organized; however, I recommend checking the grammar of the document. On lines 45 to 48 must be describe sections instead of chapters. On line 224, Fig. 9 is mentioned, however, this figure does not correspond to the described text.
Major Changes:
Table 1, only high levels of PD are analyzed, all patients in the experimental tests have levels from 2 to 4, no evidence is shown for patients of level 1, and this level is usually the most difficult to detect with respect to a healthy patient, due to the Tremor is minor. Authors must include measurements from patients with level 1 of PD in their experimental tests. Lines 186 and 187, the range of ages between patients with PD and the control group is large, to ensure reliability in the test, it is recommended to reduce this difference. For the 42 subjects, what is the distribution of valid measurements? … Due to there are 328 valid measurements and more valid tests for healthy subjects than for patients with PD, what was the selection criteria to consider only some of the tests instead of all? Lines 266 and 267, it is necessary to specify the parameters of the classifiers, for instance, the type of kernel in SVM, the number k for the K-NN, the training algorithm for the DT and NB. Line 267, why 5-fold is used during the validation process? Lines 271 to 273, it is important to mention the interest class for the binary classifier, that is, the healthy or the patient with PD? Why not use a Boosting algorithm (Random forest, AdaBoots, …), if the best result is DT? It is necessary to include a comparison table with similar works for tremor detection, I recommend including at least the follow information: Device: Rehapiano, Wearable Sensors, Leap Motion Device, smartphone,… Type of scale: FTMTRS, UPDRS,… Performance measures: accuracy, precision, recall, F1, … I suggest considering the following works in the comparison: Hssayeni, M. D., Jimenez-Shahed, J., Burack, M. A., & Ghoraani, B. (2019). Wearable Sensors for Estimation of Parkinsonian Tremor Severity during Free Body Movements. Sensors, 19(19), 4215. Vivar-Estudillo, G., Ibarra-Manzano, M. A., & Almanza-Ojeda, D. L. (2018, October). Tremor Signal Analysis for Parkinson’s Disease Detection Using Leap Motion Device. In Mexican International Conference on Artificial Intelligence (pp. 342-353). Springer, Cham. Manzanera, O. M., Elting, J. W., van der Hoeven, J. H., & Maurits, N. M. (2016). Tremor detection using parametric and non-parametric spectral estimation methods: A comparison with clinical assessment. PloS one, 11(6), e0156822. Kostikis, N., Hristu-Varsakelis, D., Arnaoutoglou, M., & Kotsavasiloglou, C. (2015). A smartphone-based tool for assessing parkinsonian hand tremor. IEEE journal of biomedical and health informatics, 19(6), 1835-1842 Include advantages and disadvantages for each proposed approach
Author Response
The paper presents a novel device to identify the action tremor in people with PD using FTMTRS. The algorithm for PD detection is interesting, however, I notice some flaws in their experimental tests and important technical errors that must be corrected.
Minor Changes:
Point 1: The paper is well organized; however, I recommend checking the grammar of the document.
Response 1: We are very grateful for your help and recommendation. We have included minor changes to our article. The English of the manuscript was corrected before submission. After our major revision, we will let it be corrected again by the MDPI English Editing Service. We include the revised manuscript and the differential manuscript in which changes are highlighted in the attachment.
Point 2: On lines 45 to 48 must be described sections instead of chapters.
Response 2: We have changed chapters to sections.
Point 3: On line 224, Fig. 9 is mentioned, however, this figure does not correspond to the described text.
Response 3: We've corrected the reference to the image.
Major Changes:
Point 4: Table 1, only high levels of PD are analyzed, all patients in the experimental tests have levels from 2 to 4, no evidence is shown for patients of level 1, and this level is usually the most difficult to detect with respect to a healthy patient, due to the Tremor is minor. Authors must include measurements from patients with level 1 of PD in their experimental tests.
Response 4: Since the first upload, we had recruited one patient that has tremor 1 and handwriting 2 rated tremor on FTMTRS. We have added the results obtained with this patient to the manuscript. Unfortunatelly, right now we don’t have access to more patients with this tremor level.
Point 5: Lines 186 and 187, the range of ages between patients with PD and the control group is large, to ensure reliability in the test, it is recommended to reduce this difference.
Response 5: Certainly decreasing the age difference between patients with PD and the control group would increase the reliability of the study but probably not much. We measure an action tremor and action tremor it’s severity has little association with age of the patient, age at PD onset, or PD duration based on [1]. We will conduct a wider range study but at this point, we had recruited a limited number of participants.
Point 6: For the 44 subjects, what is the distribution of valid measurements? Due to there are 328 valid measurements and more valid tests for healthy subjects than for patients with PD, what was the selection criteria to consider only some of the tests instead of all?
Response 6: The distribution of valid measurements of patients is uniform. When testing Hypothesis 2, we have removed all the thumb measurements (2 per session, total 98) due to a design issue (Some patients were not able to maintain contact of their thumb with the strain gauge. We need to add a fixture to improve this issue.) This is described in the paper.
Point 7: Lines 266 and 267, it is necessary to specify the parameters of the classifiers, for instance, the type of kernel in SVM, the number k for the K-NN, the training algorithm for the DT and NB.
Response 7: Thanks for the remark. We have specified the main parameters of each classifier.
Point 8: Line 267, why 5-fold is used during the validation process?
Response 8: We experimented with different values of K in K-fold cross validation. We started with 5-fold and the results with higher values of K were not significantly different. Therefore we reported our results with 5-fold cross validation.
Point 9: Lines 271 to 273, it is important to mention the interest class for the binary classifier, that is, the healthy or the patient with PD?
Response 9: The interest class is PD patient - we use this value to calculate sensitivity. We have clarified that in the paper.
Point 10: Why not use a Boosting algorithm (Random forest, AdaBoots, …), if the best result is DT?
Response 10: Our classifiers have sufficient accuracy, sensitivity and specificity to demonstrate the abilities of Rehapiano. We do not think it is necessary to use more complex classifiers for simple binary classification. However, in future in experiments to regress UPDRS, we will consider this.
Point 11: It is necessary to include a comparison table with similar works for tremor detection, I recommend including at least the follow information: Device: Rehapiano, Wearable Sensors, Leap Motion Device, smartphone,… Type of scale: FTMTRS, UPDRS,… Performance measures: accuracy, precision, recall, F1, … I suggest considering the following works in the comparison:
Hssayeni, M. D., Jimenez-Shahed, J., Burack, M. A., & Ghoraani, B. (2019). Wearable Sensors for Estimation of Parkinsonian Tremor Severity during Free Body Movements. Sensors, 19(19), 4215.
Vivar-Estudillo, G., Ibarra-Manzano, M. A., & Almanza-Ojeda, D. L. (2018, October). Tremor Signal Analysis for Parkinson’s Disease Detection Using Leap Motion Device. In Mexican International Conference on Artificial Intelligence (pp. 342-353). Springer,
Cham. Manzanera, O. M., Elting, J. W., van der Hoeven, J. H., & Maurits, N. M. (2016). Tremor detection using parametric and non-parametric spectral estimation methods: A comparison with clinical assessment. PloS one, 11(6), e0156822.
Kostikis, N., Hristu-Varsakelis, D., Arnaoutoglou, M., & Kotsavasiloglou, C. (2015). A smartphone-based tool for assessing parkinsonian hand tremor. IEEE journal of biomedical and health informatics, 19(6), 1835-1842 Include advantages and disadvantages for each proposed approach
Response 11: This is an excellent suggestion, we have incorporated the comparison table of our classifier with similar works for tremor detection into our article, please see. Table 4.
[1] Louis, Elan D., et al. "Clinical correlates of action tremor in Parkinson disease." Archives of neurology 58.10 (2001): 1630-1634.

Reviewer 5 Report
Authors developed a device - Rehapiano - for fast and quantitative assessment of action tremor. This article describes the instrumentation and results from three tests to confirm instrument validity: 1) validity against known simulated frequency generator; 2) validity by comparing healthy patients and patients with Parkinson’s
Disease, and 3) repeatability of quantitative assessment. Authors claim the Rehapiano is able to detect force changes confirmed that this system is capable of measuring action tremor. Rehapiano was also sensitive enough to enable the quantification of Parkinsonian tremor.
In general, the instrument is carefully engineered, and the authors conduct thorough validation tests of the Rehapiano. The manuscript itself is well written. For the most part, the results of hypothesis testing utilizing a relatively large sample of PD patients is impressive; however, there are some conceptual and methodologic issues that should be addressed in a revision. These are outlined below.
Conceptual Issues
While the instrumentation was designed to quantify action/postural tremor, their claim to validity came from assessments of patients with Parkinson’s disease (PD). PD is primarily a resting tremor and it’s unlikely that the Rehapiano will have clinical or research value in PD. No data were presented from patients with diagnosed action tremor, such as those with cerebellar disease or postural tremor. More importantly, the instrument as designed does not appear to permit assessment of action tremor, but only postural tremor. The authors include a very nice review of pathological tremors underscoring the etiologic and phenomenological distinctions between resting, postural, and action tremor, but the instrument seems to capture only postural tremor. The reader is left to assume that the PD patients also had postural tremor. While this may be the case for many, it is the resting tremor that first signals disease onset and is used to monitor progress or response to treatment. Any claim that the device offers clinical value needs to demonstrate that the resting tremor in PD can be detected using a force transducer. At present this is counterintuitive. A more convincing argument can be made if the authors were to report results from testing Hypothesis 3 using correlational analyses or at least a scatterplot showing strong relationships between clinical severity of resting and postural tremor and the performance of the Rehapiano. The authors claim that the Rehapiano has advantages over alternative instruments that use accelerometers (or other sensors) to record limb tremor, citing that it dose not require a sensor to be attached to the subject. Ambulatory/wearable wireless sensors actually offer significant advantages over the Rehapiano because they permit assessments in more naturalistic conditions and can be used outside the laboratory or clinic (i.e. home environment). While this strain gauge technology is indeed impressive, it actually places restrictions on the assessment protocol and is less user friendly than may devices now available on the market.Methodologic Issues
Regarding the low sensitivity to oscillations below 3 Hz, this is a critical problem. The authors assume this was due to the low torque of the DC motor at low voltage (driving the lower oscillations) causing the rubber band when stretched to slows down the motor and distort the measurement, rendering it unsuitable for low frequency tremors. The problem here is that action tremor is characterized by low frequency oscillations generally around 3 Hz. The title of the article actually refers to action tremor and this is misleading. Recommend revising the title to more accurately reflect that the instrument detects postural tremors. It is never stated in the article that the padded splints instrumented with strain gauges are completely rigid and cannot be deformed so that the applied forces that are being measured are truly isometric. There is also some concern about compatibility should others want to incorporate the Rehapiano into their clinical/research armamentarium. Specifically, serial (RS232) ports are becoming obsolete and replaced with USB interface or even wireless Bluetooth technologies. The authors might want to consider compatibility with other computers unless they view this is a complete turn-key set-up. The later will significantly reduce broader appeal and implementation of the Rehapiano in the clinical setting. Regarding the use of targeted force to standardize the assessment procedure across subjects, a decision was made to select 300gms as the target for each finger. While the authors claim that this level of force represents 2/3 of the average for all fingers, it is likely well outside the functional range for some individuals. Would it not be preferable to select a more individualized target range that represents a percentage of a subject’s maximum voluntary contraction (e.g. 20%)? In this way, force level remains standardized, but also tailored to the individual’s capability (and by extension, functional range). It is not clear what the values under the column labeled “Threshold” are. The term “threshold” is never defined. Was this just an arbitrary value used to consider the power density valid?Other issues
The investigators appeared to follow appropriate local ethical and regulatory guidelines for use of human subjects in research, as stated in the following on page 5: “We have complied with all legal and legislative conditions, including GDPR patient protection. Each patient was personally acquainted with the use of their data from the device and signed an agreement permitting processing the medical data.” However, personal identifiers (e.g. name) are presented in the data sets in the Supplementary file. Unless specifically permitted by the subject when they signed the agreement to participate, use of personal identifiers violates fundamental international research codes of ethics to protect confidentiality of human subjects. Authors should code subject identifier to protect confidentiality.
Author Response
Authors developed a device - Rehapiano - for fast and quantitative assessment of action tremor. This article describes the instrumentation and results from three tests to confirm instrument validity: 1) validity against known simulated frequency generator; 2) validity by comparing healthy patients and patients with Parkinson’s
Disease and 3) repeatability of quantitative assessment. Authors claim the Rehapiano is able to detect force changes confirmed that this system is capable of measuring action tremor. Rehapiano was also sensitive enough to enable the quantification of Parkinsonian tremor.
In general, the instrument is carefully engineered, and the authors conduct thorough validation tests of the Rehapiano. The manuscript itself is well written. For the most part, the results of hypothesis testing utilizing a relatively large sample of PD patients is impressive; however, there are some conceptual and methodologic issues that should be addressed in a revision. These are outlined below.
Conceptual Issues
Point 1: While the instrumentation was designed to quantify action/postural tremor, their claim to validity came from assessments of patients with Parkinson’s disease (PD). PD is primarily a resting tremor and it’s unlikely that the Rehapiano will have clinical or research value in PD. No data were presented from patients with diagnosed action tremor, such as those with cerebellar disease or postural tremor. More importantly, the instrument as designed does not appear to permit assessment of action tremor, but only postural tremor. The authors include a very nice review of pathological tremors underscoring the etiologic and phenomenological distinctions between resting, postural, and action tremor, but the instrument seems to capture only postural tremor. The reader is left to assume that the PD patients also had postural tremor. While this may be the case for many, it is the resting tremor that first signals disease onset and is used to monitor progress or response to treatment. Any claim that the device offers clinical value needs to demonstrate that the resting tremor in PD can be detected using a force transducer. At present this is counterintuitive. A more convincing argument can be made if the authors were to report results from testing Hypothesis 3 using correlational analyses or at least a scatterplot showing strong relationships between clinical severity of resting and postural tremor and the performance of the Rehapiano.
Response 1: We are very grateful for your help in correcting our manuscript. We tried to incorporate almost all the comments you sent us. According to [1] the prevalence of action tremor in Parkinson's disease is as high as 92%. All the PD patients in our study have been previously diagnosed using conventional methods. Their physicians reported to us that the PD patients also manifest an action tremor. This was a condition for participation in the testing. Action tremor in Parkinson's disease manifests in three forms: postural, kinetic, and isometric. We agree that it is the resting tremor that first signals PD, but we also agree with the physicians that they need to monitor the action tremor and its development too. The Rehapiano provides the doctor's insight into the development of the disease during treatment. Frequency and amplitude changes can be seen. In the future we are going to do the following comparison of rest tremor and action tremor, but we have not had the opportunity to have our own device to measure resting tremor in PD. Therefore at this point, we include reference to a publication that shows a strong correlation between resting and action tremor [1].
Point 2: The authors claim that the Rehapiano has advantages over alternative instruments that use accelerometers (or other sensors) to record limb tremor, citing that it does not require a sensor to be attached to the subject. Ambulatory/wearable wireless sensors actually offer significant advantages over the Rehapiano because they permit assessments in more naturalistic conditions and can be used outside the laboratory or clinic (i.e. home environment). While this strain gauge technology is indeed impressive, it actually places restrictions on the assessment protocol and is less user-friendly than may devices now available on the market.
Response 2: We have reconsidered our claims and we reformulated the advantages of Rehapiano. Indeed, Rehapiano is suited more for in-laboratory testing. Since the inception the device was intended not only for diagnostics but also for rehabilitation of motoric impairments (not related solely to PD). Regarding PD diagnostics and assessment, we consider that the advantage of Rehapiano is that it provides a very controlled setup for measurement of action isometric tremor, that has not been studied much.
Methodologic Issues
Point 3: Regarding the low sensitivity to oscillations below 3 Hz, this is a critical problem. The authors assume this was due to the low torque of the DC motor at low voltage (driving the lower oscillations) causing the rubber band when stretched to slow down the motor and distort the measurement, rendering it unsuitable for low-frequency tremors. The problem here is that action tremor is characterized by low-frequency oscillations generally around 3 Hz.
Response 3: We have probably failed to explain that we don’t claim that our device has lower sensitivity do oscillations below 3Hz, just that we did not verify it’s sensitivity because of limitations of the actuator and the rubber band that we have used. We have changed the formulations accordingly and we hope that it will be clearer now. In the future, we will prepare a new verification experiment for the lower frequency oscillations. According to our measurements and according to [2] the range of action tremor is from 3.5 Hz–7 Hz, therefore by all respect we don’t consider not measuring lower frequencies as a critical flaw of the study.
Point 4: The title of the article actually refers to action tremor and this is misleading. Recommend revising the title to more accurately reflect that the instrument detects postural tremors. It is never stated in the article that the padded splints instrumented with strain gauges are completely rigid and cannot be deformed so that the applied forces that are being measured are truly isometric.
Response 4: When referring to various kinds of tremor we use the terminology of the following publication “Consensus Statement of the Movement Disorder Society on Tremor” (Günther Deuschl MD Peter Bain Ma, MD,Frcp Mitchell Brin MD Ad Hoc Scientific Committee, https://onlinelibrary.wiley.com/doi/epdf/10.1002/mds.870131303). The authors state that there are two basic kinds of tremor, rest tremor, and action tremor. Quote:”Action tremor is any tremor that is produced by voluntary contraction of muscle, including postural, isometric, and kinetic tremor. The latter includes intention tremor... Postural tremor is present while voluntarily maintaining a position against gravity...Isometric tremor Tremor occurring as a result of muscle contraction against a rigid stationary object (for example, while making a fist or squeezing the examiner's fingers).” Based on this division we consider that we measure action tremor, namely isometric tremor and consider the title of our paper to reflect it correctly. We added that the hands are in a fixed position during exercise.
Point 5: There is also some concern about compatibility should others want to incorporate the Rehapiano into their clinical/research armamentarium. Specifically, serial (RS232) ports are becoming obsolete and replaced with USB interface or even wireless Bluetooth technologies. The authors might want to consider compatibility with other computers unless they view this is a complete turn-key set-up. The later will significantly reduce broader appeal and implementation of the Rehapiano in the clinical setting.
Response 5: The Rehapiano uses a Universal Serial Bus. We had a mistake in our manuscript and we have corrected it. We have used a serial link connection over the USB interface. We have also changed the scheme in Figure 3. Our computer also does not feature an RS232 port.
Point 6: Regarding the use of targeted force to standardize the assessment procedure across subjects, a decision was made to select 300gms as the target for each finger. While the authors claim that this level of force represents 2/3 of the average for all fingers, it is likely well outside the functional range for some individuals. Would it not be preferable to select a more individualized target range that represents a percentage of a subject’s maximum voluntary contraction (e.g. 20%)? In this way, force level remains standardized, but also tailored to the individual’s capability (and by extension, functional range).
Response 6: This is an excellent idea. During the initial measurements, we determined the value of 300gms as written. However, we plan to change the setpoint setting for each individual as you recommend.
Point 7: It is not clear what the values under the column labeled “Threshold” are. The term “threshold” is never defined. Was this just an arbitrary value used to consider the power density valid?
Response 7: We have added the explanation of the peak power proportion threshold below Table 5.
Point 8: The investigators appeared to follow appropriate local ethical and regulatory guidelines for use of human subjects in research, as stated in the following on page 5: “We have complied with all legal and legislative conditions, including GDPR patient protection. Each patient was personally acquainted with the use of their data from the device and signed an agreement permitting processing the medical data.” However, personal identifiers (e.g. name) are presented in the data sets in the Supplementary file. Unless specifically permitted by the subject when they signed the agreement to participate, use of personal identifiers violates fundamental international research codes of ethics to protect the confidentiality of human subjects. The authors should code the subject identifier to protect confidentiality.
Response 8: Thank you for pointing it out. We have changed the names of the Rehapiano participants into 6-digit codes. We have also removed the names info files. We will upload a new supplementary file with altered patient coding.
General remark: The English of the manuscript was corrected before submission. After our major revision, we will let it be corrected again with the MDPI English Editing Service. We include the manuscript after editing and the differential manuscript in which changes are better visible in the attachment.
[1] Parkinson tremor have action tremor. Clinical Correlates of Action Tremor in Parkinson's Disease Elan D. Louis, MD, MS; Gilberto Levy, MD; Lucien J. Côte, MD; et al
[2] Lee, Hong Ji, et al. "Tremor frequency characteristics in Parkinson's disease under resting-state and stress-state conditions." Journal of the neurological sciences 362 (2016): 272-277.

Round 2
Reviewer 1 Report
The authors have considerably improved the manuscript. In my opinion, it is acceptable for publication in the journal Sensors after a few more minor improvements.
The are still misspelling and grammar problems. A few of them are mentioned below:
Page 2, line 39: “The authors” instead of “Authors”
Page 6, line 202: “Sessions” instead of “Sesions”
Page 6, line 195: “The force is visualized by a green bar.”
I would suggest: The amplitude of this force is displayed using vertical bars in Fig. 5.
Page 11, line 305: missing a “the” prior to “weighted average”.
Some changes remain unclear to me. These are a few:
Page 6, Table 1: After the inclusion PD patient 7, tremor for patient 2 and handwriting for patients 3 and 4 were changed without the inclusion of new sessions. For these last two, it seems that a new assumption was included, and these last two patients were re-rated. What is the real reason of this changes?
Page 11, line 195: How was the F1-scores calculated?
Author Response
Point 1:
The authors have considerably improved the manuscript. In my opinion, it is acceptable for publication in the journal Sensors after a few more minor improvements.
Response 1:
Dear reviewer,
We are again very grateful for your help in correcting our manuscript. We appreciate your time spent correcting our article. Our article has been fixed by MDPI English Editing Service, following your recommendation.
Point 2:
The are still misspelling and grammar problems. A few of them are mentioned below:
Page 2, line 39: “The authors” instead of “Authors”
Page 6, line 202: “Sessions” instead of “Sesions”
Page 6, line 195: “The force is visualized by a green bar.”
I would suggest: The amplitude of this force is displayed using vertical bars in Fig. 5.
Page 11, line 305: missing a “the” prior to “weighted average”.
Response 2:
We have included minor grammar changes.
Point 3:
Some changes remain unclear to me. These are a few:
Page 6, Table 1: After the inclusion PD patient 7, tremor for patient 2 and handwriting for patients 3 and 4 were changed without the inclusion of new sessions. For these last two, it seems that a new assumption was included, and these last two patients were re-rated. What is the real reason of this changes?
Response 3:
The real reason for these changes is just the application we used to compare old and modified manuscript. We added patient to the manuscript. However, all patients moved a row and an application that compared the table in the old and new manuscripts gave the impression that the patient's data had been changed. Patient data was not changed, only in the difference set it looked like that.
Point 4:
Page 11, line 195: How was the F1-scores calculated?
Response 4:
We have added the formula for calculating F1 and added the precision column to the table. We also found an error in calculating the F1 score for NB, which we corrected.

Reviewer 2 Report
After reconsideration for publication by the journal, my initial remarks have been adressed. Therefore, I support publication in current form, with minor english revision by the publisher
Author Response
Point 1: After reconsideration for publication by the journal, my initial remarks have been adressed. Therefore, I support publication in current form, with minor english revision by the publisher.
Response 1:
Dear reviewer,
We are again very grateful for your help in correcting our manuscript. We appreciate your time spent correcting our article. Our article has been fixed by MDPI English Editing Service, following your recommendation.

Reviewer 4 Report
I consider that the current version of the paper complies with most of my observations regarding the previous version, clarifying the points that due to time it is not possible to resolve
Author Response
Point 1: I consider that the current version of the paper complies with most of my observations regarding the previous version, clarifying the points that due to time it is not possible to resolve.
Response 1:
Dear reviewer,
We are again very grateful for your help in correcting our manuscript. We appreciate your time spent correcting our article. Our article has been fixed by MDPI English Editing Service, following your recommendation.

Reviewer 5 Report
authors adequately addressed prior concerns; revision included stronger support for design considerations and where necessary, authors corrected errors and/or misleading statements or conclusions. The authors should be commended for their thoughtful responses. I have no remaining issues or concerns with this fine technical manuscript.
Author Response
Point 1: authors adequately addressed prior concerns; revision included stronger support for design considerations and where necessary, authors corrected errors and/or misleading statements or conclusions. The authors should be commended for their thoughtful responses. I have no remaining issues or concerns with this fine technical manuscript.
Response 1:
Dear reviewer,
We are again very grateful for your help in correcting our manuscript. We appreciate your time spent correcting our article. Our article has been fixed by MDPI English Editing Service, following your recommendation.
